# Effects of Combined Movement and Storytelling Intervention on Fundamental Motor Skills, Language Development and Physical Activity Level in Children Aged 3 to 6 Years: Study Protocol for a Randomized Controlled Trial

**DOI:** 10.3390/children10091530

**Published:** 2023-09-09

**Authors:** Rodrigo Vargas-Vitoria, César Faúndez-Casanova, Alberto Cruz-Flores, Jordan Hernandez-Martinez, Stefany Jarpa-Preisler, Natalia Villar-Cavieres, María Teresa González-Muzzio, Lorena Garrido-González, Jorge Flández-Valderrama, Pablo Valdés-Badilla

**Affiliations:** 1Department of Physical Activity Sciences, Faculty of Education Sciences, Universidad Católica del Maule, Talca 3530000, Chile; rvargas@ucm.cl (R.V.-V.); cfaundez@ucm.cl (C.F.-C.); 2Doctorado en Educación en Consorcio, Facultad de Ciencias de la Educación, Universidad Católica del Maule, Talca 3480094, Chile; a.ignaciocruz@gmail.com; 3Department of Physical Activity Sciences, Universidad de Los Lagos, Osorno 5290000, Chile; jordan.hernandez@ulagos.cl; 4Programa de Investigación en Deporte, Sociedad y Buen Vivir, Universidad de Los Lagos, Osorno 5290000, Chile; 5Carrera de Medicina, Facultad de Medicina, Universidad Católica del Maule, Talca 3480094, Chile; stefanyjarpapreisler@gmail.com; 6Departamento Formación Inicial Escolar, Facultad de Ciencias de la Educación, Universidad Católica del Maule, Talca 3480094, Chile; nvillar@ucm.cl (N.V.-C.); mtgonzal@ucm.cl (M.T.G.-M.); lgarridog@ucm.cl (L.G.-G.); 7Escuela de Educación Parvularia, Facultad de Ciencias de la Educación, Universidad Católica del Maule, Talca 3480094, Chile; 8Escuela de Educación Física, Deportes y Recreación, Instituto de Ciencias de la Educación, Universidad Austral de Chile, Valdivia 509000, Chile; jflandez@uach.cl; 9Sports Coach Career, School of Education, Universidad Viña del Mar, Viña del Mar 2520000, Chile

**Keywords:** motor skills, language acquisition, exercise, preschool, child

## Abstract

This study protocol aims to analyze and compare the effects of combined movement and storytelling intervention (CMSI) on fundamental motor skills (locomotor skills and object control), language development (language comprehension, language expression, vocabulary and language description), and physical activity levels (light intensity, moderate-to-vigorous intensity and sedentary time) in children aged 3 to 6 years. The sample will consist of 144 children from 12 class groups, randomly assigned to 3 experimental groups (*n* = 72 children) and 3 control groups (*n* = 72 children), belonging to 4 class groups of upper-middle-level classes (2 experimental and 2 control; 3 to 4 years), 4 transition level 1 classes (2 experimental and 2 control; 4 to 5 years) and 4 transition level 2 classes (2 experimental and 2 control; 5 to 6 years). The experimental groups will perform CMSI for 3 sessions per week (40 min per session) over 12 weeks (using one motor story per week), while the control groups will not receive any treatment. The main outcome will provide information about fundamental motor skills, language development, and physical activity levels. Our hypothesis indicates that CMSI has the potential to generate significant increases in selected assessments. If this intervention proves to be beneficial, it could contribute to preschool and school curricula.

## 1. Introduction

A physically active lifestyle should begin at an early age and continue throughout the life cycle [1,2], with a synergistic relationship between physical activity and motor competence [3]. The development of fundamental motor skills (FMSs) is a favorable antecedent and significantly related to longer periods of moderate-to-vigorous physical activity (MVPA) in children (*p* < 0.05) [4]. FMSs denote the mastery of common basic motor skills with specific patterns; these can be classified as locomotor, object control, and stability skills, which favor the development of advanced and complex movements that allow participation in high-motor-engagement activities [4]. 

Thus, FMSs represent a key factor for healthy development in early childhood [5,6]. A systematic review conducted by Bolger et al. [7] showed that FMS competence increases with age during childhood, with locomotor skills being greater than object control. In addition, physical inactivity leads to impaired motor competence and correlates negatively with language development (β = −0.664; *p* = 0.004) in children aged 3 to 5 years [8]. Furthermore, Wang et al. [9] indicated that stability for the motor domain within each developmental area is high (motor competence) from 3 to 5 years of age, while unique prediction from one domain to the other is weak.

In this regard, a systematic review by Lin et al. [10] reported that FMS-focused physical activity interventions in children lead to improved postural control, motor competence and muscle strength compared to traditional physical education classes. In a meta-analysis conducted by Van Capelle et al. [11], FMS-focused interventions lasting 17 to 21 weeks with a frequency of 1 to 3 sessions of 17 to 35 min, led by teachers, significantly improved overall FMSs (SMD = 0.14; 95% CI = 0.06 to 0.21; *p* = 0.0003), object control (SMD = 0.47; 95% CI = 0.15 to 0.80; *p* = 0.004) and locomotor skills (SMD = 0.44; 95% CI = 0.16 to 0.73; *p* = 0.002) compared to children aged 3 to 5 years who were given no teacher-led FMS interventions. In another study, Han et al. [12] reported a moderate correlation between FMS scores in executive functions (r = 0.33; *p* < 0.001) and scores in locomotor skills, which were found to be significant predictors of inhibitory control (β = 0.21; *p* < 0.001), working memory (β = 0.18; *p* < 0.01) and cognitive flexibility (β = 0.24, *p* < 0.001) in children aged 3 to 5 years. 

Additionally, a systematic review by Jylänki et al. [13] reported that FMS interventions were effective in improving executive functions, language, arithmetic skills, cognitive skills, and memory in children aged 3 to 7 years; however, these studies presented a low methodological quality. In general, FMS proposals are based on locomotor skills, object control, and stability skills, with less common FMS strategies involving not only motor competence but also relating to motor stories that are used to stimulate the cognitive state of schoolchildren [14]. 

Combined movement and storytelling intervention (CMSI) is emerging as an FMS strategy that has shown significant improvements in motor, cognitive and linguistic skills in children [14,15]. In a study conducted by Eyre et al. [15], involving a non-randomized controlled trial with a 12-week CMSI, significant improvements (*p* ≤ 0.01) were reported in seven motor skills (run, jump, throw, catch, stationary dribble, roll and kick) in South Asian children aged 5 to 6 years. Similarly, Duncan, Cunningham and Eyre [14] conducted a randomized controlled trial over 6 weeks, using CMSI with preschool children in England, and reported significant improvements in motor (Δ 4.86; *p* = 0.001) and language (Δ 13.6; *p* = 0.001) skills compared to two active control groups. 

Taking into account the benefits of promoting FMSs at an early age and considering the dearth of scientific literature to date on novel strategies such as CMSI [14,15], it becomes relevant to analyze the possible changes/benefits of introducing CMSIs to preschool children. Therefore, this study protocol aims to analyze and compare the effects of CMSI on FMSs (locomotor skills and object control), language development (language comprehension, language expression, vocabulary and language description) and physical activity levels (light physical activity, LPA; MVPA; and sedentary behavior, SB)—in children aged 3 to 6 years. Based on previous studies [14,15], it is hypothesized that CMSI has the potential to generate significant increases in locomotor skills, object control, language comprehension, language expression, vocabulary, language description and MVPA level compared to control groups.

## 2. Material and Methods

### 2.1. Study Design

The study is based on a double-blind, repeated measures and parallel groups (6 interventions and 6 controls) (randomized controlled trial) experimental design; considering previous studies, a quantitative approach will be adopted [16]. The methodology followed will be the Consolidated Standards of Reporting Trials Statement (CONSORT) methodology [17]. 

### 2.2. Ethical Approval

The current protocol has been reviewed and approved by the Scientific Ethics Committee of the Universidad Católica del Maule, Chile (approval number: N°105/2021, 4 August 2021) and developed following the Declaration of Helsinki for work with humans. In addition, it has been registered in the Clinical Trial Protocol Registry and Results System (ClinicalTrials.gov (https://clinicaltrials.gov/ct2/home, accessed on 28 August 2022) of the United States of America (code: NCT06016296).

### 2.3. Sample Size Calculation

Twelve children will be recruited from each class group (without gender distinction). The distribution of the class groups (12 class groups) will be in 3 experimental groups (6 class groups, n ± 72 children) and 3 control groups (6 class groups, n ± 72 children). The distribution by clusters or strata is justified by the fact that the use of individual randomization would imply a high probability of contaminating the control groups since it is unfeasible to prohibit children from interacting in school classrooms or recreational spaces (i.e., playgrounds, hallways, laboratories), which is where the study intervention will be carried out [18,19]. In addition, in the Maule region (one of the sixteen regions into which to Chile is divided; located in the center of the country and bordered to the north by the O’Higgins region, to the east by the provinces of Mendoza and Neuquén belonging to Argentina, to the south by the Ñuble region and to the west by the Pacific Ocean), there are 54 educational establishments that provide preschool education with an enrollment of approximately 1080 children. 

The sample size calculation indicates that the ideal number of participants per group is 10. As agreed in a previous study by Eyre et al. [15], we will use a mean difference of 3.22 total score of locomotor skill domains for this calculation as the minimum difference necessary for substantial clinical relevance, with a standard deviation of 0.70 points, considering an alpha level of 0.05 with a power of 80% and an expected loss of 15%. The GPower program (version 3.1.9.6, Franz Faul, Universiät Kiel, Kiel, Germany) will be used to calculate the statistical power. 

### 2.4. Randomization and Blinding

The 12 class groups will be selected from 4 upper-middle-level classes (2 experimental and 2 control; age range between 3 and 4 years), 4 transition level 1 classes (2 experimental and 2 control; age range between 4 and 5 years) and 4 transition level 2 classes (2 experimental and 2 control; age range between 5 and 6 years), which will be randomized with stratified sampling [16]. This consists of segmenting the classes that agree to participate in the study according to educational levels (strata) and then performing a random sampling on each of them, using R statistical software, version 4.1.2. This study is considered double-blind because the measurements will be performed by professionals who are external to the research. 

### 2.5. Participants

We aim to enroll 144 children residing in the Maule region, Chile, who meet the following inclusion criteria: (i) enrolled in an educational center (school, college or kindergarten) that agrees to participate in the intervention; (ii) age range between 3 and 6 years old; (iii) attend ≥ 85% of the sessions scheduled for CMSI. Exclusion criteria will be the following: (i) children with musculoskeletal injuries or medical contraindications (i.e., congenital heart disease, fever, diarrhea or general malaise) that would prevent their normal performance in the assessments and intervention; and (ii) children with permanent educational needs mentioned in Decree No.83 of the Chilean Ministry of Education [20], such as visual, hearing, intellectual or multiple disabilities, dysphasia or autistic disorder. Figure 1 summarizes the selection process of the class groups and the number of children who will be included in the study.

### 2.6. Intervention

The intervention will be implemented by early childhood educators, who will be trained about the objectives and contents of each session of CMSI that will be developed. Each session will take place in the classroom the preschoolers usually attend, which will depend on the admission policies of each educational community regarding the class levels (upper middle, transition level 1, and transition level 2). The intervention will last 12 weeks (36 sessions), distributed in 3 weekly sessions (Monday, Wednesday and Friday) of 40 min per session. CMSI is based on 2 previous studies [14,21], and a book related to CMSI [22]. Twelve unpublished motor stories will be distributed, one each week over 3 sessions, with the corresponding progressions for each session carried out. Thus, the children will be placed in a sequence of stories that begin with the presentation of a motivating character, who will accompany the children during the development of the whole story. The motor stories that will be developed each week are: (i) who I am; (ii) we go on an excursion; (iii) the school; (iv) the recess; (v) the weekend has arrived; (vi) visiting the park; (vii) my days in the field; (viii) my days at the beach; (ix) we go to the stadium; (x) we are going to know my environment; (xi) let’s go to the circus; (xii) tell me your adventures. The aim of the motor story translates into solving practical challenges while maintaining control, balance and coordination by combining various postures and movements such as throwing and receiving, moving on inclined planes and following rhythms, all in a variety of games and other activities [4,12,23].

The 40 min sessions will include three segments: (i) warm-up (5 min), which consists of joint mobility exercises and introduction to the motor story of the week; (ii) main part (25 min), which incorporates the narration of the motor story by the educator initially, as well as the participation of the children through their own corporeality with movements based mainly on FMSs (locomotion skills and object control), through two-way verbal interaction based on questions relating to the narration delivered by the educator or through direct verbal participation from each child; and (iii) cool down (10 min), where space is provided for the children to gradually return to calm, together with feedback on the content addressed during the session. Table 1 summarizes the materials and objectives of the weekly motor stories.

### 2.7. Control Groups

The upper-middle-level control groups (n = 2 classes; n = 24 children), transition level 1 control groups (n = 2 classes; n = 24 children) and transition level 2 control groups (n = 2; n = 24 children) will participate in the assessments (initial and final) and will be asked to maintain their regular sessions at their educational establishments. At the end of the intervention period (according to the results obtained), the control group children, parents and/or legal guardians, managers and coordinators of each community will be contacted and provided with the material so that they can replicate the experience. In addition, the educators of these establishments (teachers and classroom assistants) will be trained in CMSI strategy.

### 2.8. Outcomes and Procedures

The first meeting will be to obtain the agreement of the educational establishments and educators, to obtain the necessary signatures of informed consent and to explain the scope and aim of the study; also, the children will be randomly assigned to the intervention groups. During the following 2 weeks, the first assessments (pre-tests) will be carried out, before the 12-week CMSI is implemented; finally, the second assessments (post-tests) will be administered. One of the researchers will supervise all of the assessments and CMSI, and there will be a predefined order that will be followed by expert personnel in the assessments: (i) FMSs (physical education teacher specializing in early childhood care); (ii) language development (kindergarten teacher); (iii) physical activity level (physical education teacher); (iv) morphological variables (anthropometrist); and (v) sociodemographic variables (kindergarten teacher and physical education teacher). Figure 2 shows a summary of the assessments and sessions that are usually part of CMSI.

### 2.9. Primary Outcomes

#### 2.9.1. Fundamental Motor Skills (FMSs)

These will be assessed by means of the test of gross motor development-second edition (TGMD-2) [24]. This battery measures 12 FMSs in two different domains (locomotor skills and object control). The locomotor skills domain contains 6 assessments: (i) running, (ii) galloping, (iii) hopping on one foot, (iv) long jumping, (v) horizontal jumping and (vi) lateral sliding. The object control domain also includes 6 assessments: (i) batting a stationary ball, (ii) stationary dribbling, (iii) catching a ball, (iv) kicking a ball, (v) throwing a ball and (vi) rolling a ball. Estimated administration time is ≤10 min per child. The 6 locomotion skills and 6 object control skills incorporate 24 criteria, allowing a total score from 0 to 48 points. The domain scores can be transformed into a standard score (median = 10.0 and standard deviation = 3.0) and subsequently into a gross motor quotient (median = 100.0 and standard deviation = 15.0). This information can be found in percentiles that allow the child to be placed in normative values with respect to a reference population. The original TGMD-2 reports high test–retest and inter-rater reliability (all r values > 0.85), as well as good internal consistency that varies between 0.85 and 0.88 for each ability [24]. In addition, its content, construct and concurrent validity have been demonstrated in children aged ≥ 3 years in countries such as Chile [25,26], Brazil [27] and Korea [28]. 

#### 2.9.2. Language Development

The language test for preschoolers (TELEPRE) [29] will be used. The purpose of this assessment is to measure the language of children in initial educational levels (3 to 6 years old) by means of 4 domains: (i) language comprehension, (ii) language expression, (iii) vocabulary and (iv) language description. The administration of the instrument requires a booklet of questions, various objects (bottle, toy car, pencil, plate, needle, cup, paintbrush, screw, sponge, small ball, button, small book, spoon, scissors, matchbox; plus three objects that serve as distractors) and 3 pictures representative of situations (i.e., serving milk to a cat, setting the table and being in a toy store) to be described by the child. To administer the instrument, a room free from noise or other distraction is required, in which the evaluator, the child and a classroom assistant will be present. The estimated application time is ≤25 min. The test scores vary according to the domain and age of the child being evaluated; they are described in detail below, according to the original manual [29]. To interpret the scores, the original manual [29] proposes the use of T-scores (average = 50 points and standard deviation = 10). Regarding the validation of the instrument, its internal consistency varies between 0.87 and 0.93, according to the domain analyzed and a concurrent validity with the test of psychomotor development (TEPSI) [29].

#### 2.9.3. Physical Activity Level

This will be objectively monitored by accelerometers (ActiGraph GT9X, Pensacola, FL, USA). The device will be worn at the waist on an elastic belt, at the mid-axillary line on the right side. Children will be instructed to wear the accelerometer 24 h a day, for at least 7 days, including 2 weekend days, and only to remove the device when bathing or engaging in water activities [30]. The minimum amount of data considered acceptable for analysis purposes will be five days (including one weekend day) [30], with at least 10 h/day of wear time. LPA, MVPA and SB, following the cut-off points proposed by Lettink et al. [31], will be considered for analysis. Sleep time will not be taken into account. Data will be verified using Actilife software version 5.6 (ActiGraph, Pensacola, FL, USA). Consecutive 20 min blocks of 0 count will be considered as non-use of the device and discarded from the analyses. Data will be collected at a sampling rate of 30 Hz, downloaded in one-second periods, and aggregated over 15-s periods [32]. 

### 2.10. Secondary Outcomes

#### 2.10.1. Morphological Variables

Bipedal height will be measured by placing a tape measure (Bodymeter 206, SECA, Hamburg, Germany; accuracy of 0.1 cm) on the wall and utilizing the Frankfort plane in a horizontal position. The body weight will be determined using an electronic scale (Tanita BC-730 Tokyo, Japan; accuracy of 0.1 kg) and the body mass index will be computed by dividing the body weight by the square of the bipedal height (kg/m^2^). Waist circumference and bicep circumference will be measured using an inextensible tape measure (Seca-201, Hamburg, Germany) with an accuracy of 0.1 cm. Similarly, skinfolds (mm) will be measured in the bicipital, tricipital, subscapular and suprailiac regions using a plicometer (Harpenden-FG1056, England; accuracy of 0.2 mm) that exerts a constant pressure of 10 g/mm^2^. These variables will be assessed by an anthropometrist, level II, certified by the International Society for the Advancement of Kinanthropometry (ISAK) [33]. Measurements will be taken during the morning of a fasting day in a conditioned room.

#### 2.10.2. Sociodemographic Variables

A questionnaire integrating items proposed in previous studies [34] will be used for this measure. Legal guardians will be asked to respond to the following: (i) child’s name; (ii) child’s date of birth; (iii) in which country was the child born; (iv) in which country I was born; (v) what is my relationship to the child; (vi) marital status of the parents or guardians; (vii) in what area does the child reside; (viii) does the place where the child lives have a yard; (ix) how many people live with the child; (x) how many persons work in your household; (xi) what is the total family income of the people living with the child (approximately); (xii) what is the highest level of schooling you have. 

### 2.11. Statistical Analysis

R statistical software, version 4.1.2, will be used [35]. Descriptive statistics will be generated where the summary and dispersion measures of the data will be calculated. The variables will be subjected to the Shapiro–Wilk test for normality and Levene’s test for homogeneity of variance.

### 2.12. Intention-to-Treat Analysis

A two-factor repeated measures ANOVA will be used to determine the effects of the intervention on the outcomes (group × time). Post hoc tests will be performed with an alpha adjusted by Bonferroni to identify statistically significant differences. The effect sizes will be calculated utilizing Cohen’s d [36] within the group and between the groups, using the following equation: effect size = (post-mean − pre-mean)/standard deviation combined. The level of significance will be set at 5%. The intraclass correlation coefficient will be used to verify the measurement’s reliability, with a predetermined threshold of 0.80 being used to include the data in the analyses. Furthermore, logistic regression will be performed to identify the association of the physical activity level with FMSs, language development and morphological and sociodemographic variable factors. For all statistical analyses, the sociodemographic variables will be considered as moderators (covariates). 

### 2.13. Analysis by Protocol

For this analysis, only children who have completed more than 85% of the sessions will be included. The same procedures will be performed as those indicated for the intention-to-treat analysis.

## 3. Discussion

As mentioned in previous studies [11,12,13], FMS interventions lead to improvements in cognitive function, basic motor skills, language and memory in children aged 3 to 5 years. However, despite CMSI showing beneficial effects on basic motor skills and language in children [14,15], to date, there have been few studies that confirm its efficacy. This study responds to several concerns and needs identified in the specialized literature on motor competence, which suggests implementing interventions from early childhood, based on a robust methodology, in a structured learning environment and with a well-defined purpose [4,13]. 

The programs using CMSI for a period of 12 weeks, with one session per week, have demonstrated improvements in FMSs such as running, jumping, throwing, catching, stationary dribbling, rolling and kicking in South Asian children aged 5 to 6 years [15]. Similarly, Duncan, Cunningham and Eyre [14] found improvements in FMSs and language development following a 6-week intervention, consisting of one session per week, when using CMSI in England with children aged 3 to 5 years. 

This background supports the design to be implemented in the present protocol, which could show positive effects on FMSs in Chilean children aged 3 to 6 years. Since the effects of CMSI in Latin American children are unclear, this protocol will identify the effects on FMSs, language development and physical activity levels in Chilean children. Scientific evidence affirms that FMSs are an important precursor to psychological, social and physical benefits from the early stages of human development [5,37,38]. It is also recognized that language development is related to FMSs and motor competence from the early stages [7,8,9]. 

In this context and in order to avoid possible biases, children with musculoskeletal injuries, medical contraindications or congenital heart disease, and children with special educational needs such as visual, hearing, intellectual or multiple disabilities, dysphasia or autistic disorder that would impede their normal physical performance in the assessments and intervention, will be excluded. 

In addition, the study protocol has the strength to consider as secondary measures a series of morphological and sociodemographic variables in children aged 3 to 6 years. Morphological changes in preschoolers have been reported to lead to better reading comprehension [39]. In another study, conducted by Ferreira et al. [40], it was reported that children aged 6 to 7 years living in homes classified as adequate, located in neighborhoods outside the central region, and studying in private schools, had increased opportunities for better motor development in FMSs. 

All these variables and their respective indicators allow for the control of potential factors that go beyond the primary outcomes. Therefore, it is important to include these measurements in the programs for CMSI in children from 3 to 6 years of age, which can provide positive effects on these variables at an early age. If they do, then these interventions should be implemented in various schools with preschool classes. 

Further advantages of proposing such forms of intervention for Chilean children from 3 to 5 years of age in kindergartens and first grade are: (i) little space is required, and the intervention can be carried out in the classroom; (ii) little implementation is needed because most of the activities are carried out individually or in groups; (iii) low costs are required to carry out this type of intervention as a large number of these materials are already in the classroom; (iv) the variety of motor stories available can support high motivation and adherence to the intervention. 

Among the possible limitations of the study protocol are: (i) difficulty of recruiting children who can adhere to the study for the full 12 weeks plus the pre-intervention and post-intervention assessment periods; (ii) some children are unable to complete CMSI sessions due to illness or injury. 

## 4. Conclusions

This study protocol will analyze and compare the effects of CMSI on FMSs (locomotor skills and object control), language development (language comprehension, language expression, vocabulary and language description) and physical activity levels (light activity, moderate-to-vigorous intensity and sedentary time) in children aged 3 to 6 years. Our hypothesis indicates that CMSI has the potential to generate significant increases in locomotor skills, object control, language comprehension, language expression, vocabulary, language description and moderate-to-vigorous intensity of physical activity levels compared to control groups. If this intervention proves to be beneficial, it could contribute to the curriculum and comply with the indicative performance standards linked to establishments that provide education to children between 3 and 6 years of age. Moreover, this study represents a significant contribution to scientific knowledge in this line of research as, to the best of our knowledge, no studies using CMSI with Latin American preschoolers have yet been published. 

## Figures and Tables

**Figure 1 children-10-01530-f001:**
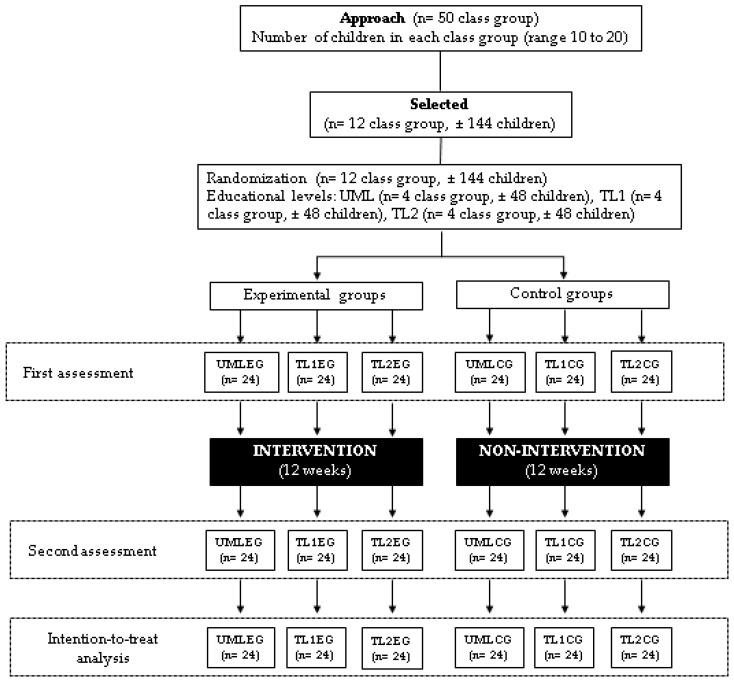
Study flowchart. Note: UMLEG: upper-middle-level experimental groups, UMLCG: upper-middle-level control groups, TL1EG: transition level 1 experimental groups, TL1CG: transition level 1 control groups, TL2EG: transition level 2 experimental groups, TL2CG: transition level 2 control groups.

**Figure 2 children-10-01530-f002:**
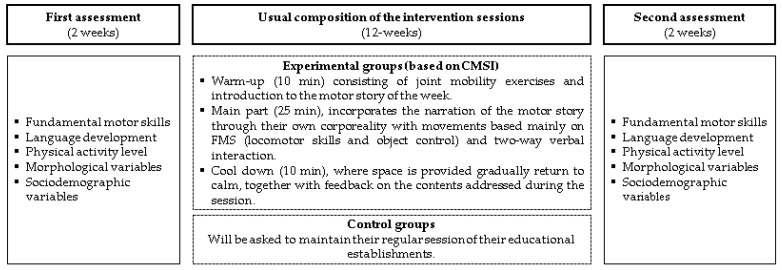
Assessments and usual sessions of the intervention. Legends: CMSI: combined movement and storytelling intervention. FMSs: fundamental motor skills.

**Table 1 children-10-01530-t001:** Summary of combined movement and storytelling intervention.

Motor Story	Materials	Warm-Up (5 min)	Main Part (25 min)	Cool Down (10 min)
Week 1: Who I am	Chairs, hoops, balls, mats, pencils, erasers and masking tape	Joint mobility exercises ^1^ and introduction to motor story	Locomotion skills ^2^ and verbal interaction ^4^	Return to calm ^5^ and feedback from the motor story
Week 2: We go on an excursion	Mats, chairs and rope		Notion and body image, and dynamic balance	
Week 3: The school	Mats, chairs, hula hoops and voice		Notion and body image, dynamic balance and visuomotor coordination	
Week 4: The recess	Mats, materials available in the patio or classroom		Dynamic balance, object control ^3^ and free play	
Week 5: The weekend has arrived	Music, blanket, mats, hoops, balls and newsprint		Object control ^3^ and verbal interaction ^4^	
Week 6: Visiting the park	Games found in the kindergarten playground and hula hoops		Dynamic balance, tone control, eye–motor coordination and object control ^3^	
Week 7: My days in the field	Lentil cones, and ping pong balls from reused paper		Dynamic balance, eye–motor coordination and free play	
Week 8: My days at the beach	Relaxing music, speaker and elements in hand		Sensorimotor, dynamic balance and visuomotor coordination	
Week 9: We go to the stadium	Exercise machines, speaker, relaxing music and relaxation audio		Dynamic balance, eye–motor coordination and free play	
Week 10: We are going to know my environment	Ball, relaxing music, speaker and voice		Dynamic balance and eye–motor coordination	
Week 11: Let’s go to the circus	Balls, ropes, mats and bottles		Locomotion skills ^2^, object control ^3^ and verbal interaction ^4^	
Week 12: Tell me your adventures	Speaker, relaxation music, sock balls and other materials available		Dynamic balance, eye–motor coordination and tone control	

Retrieved from Conde [22] and adapted by authors. ^1^ Included joint movements of the lower and upper limbs. ^2^ Included running, jumping, galloping, rolling and crawling, among others. ^3^ Included throwing, catching and kicking, among others. ^4^ For example, the teacher asks what animals live in the jungle and the children answer orally and then imitate some of them, through body movements. ^5^ Breathing exercises and stretching of the muscles involved in the activities.

## Data Availability

Not applicable.

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
