# Peer review of "Effects of Combined Movement and Storytelling Intervention on Fundamental Motor Skills, Language Development and Physical Activity Level in Children Aged 3 to 6 Years: Study Protocol for a Randomized Controlled Trial"

_children, 2023, doi:10.3390/children10091530_

Round 1

Reviewer 1 Report

The purpose of this study was to analyze and compare the effects of a 12-week combined motor and storytelling intervention on basic motor skills, language development, and physical activity levels of 3 - 6 years old children. It makes sense for research to make explorations of basic motor skill development and physical activity promotion in preschoolers. However, there are still some areas for improvement.

First of all, as with all papers, the design of a PROTOCOL paper requires the two essential elements of innovation and rigor. In terms of implementation, the STUDY PROTOCOL is even more rigorous. For example, clinical trial registration is an essential part of protocol submissions, especially for interventional clinical trials, which must be registered before the trial can begin. Successful registration with Prospero results in a CRD-header number exclusive to the study, which authors are asked to add to this article.

Secondly, the methodology section proposes to add the following clarification.

(1) How were subjects recruited? Were all students given a permission slip and then only the students' whose parents signed the permission slip were included in the study? Do children on the way to intervention have the right to withdraw from the experiment?

(2) In 2. 7, please add what activities the control group students were doing during the 12-week intervention experiment? I would like to know what the control group is doing when the intervention group is performing movement exercises 3 times a week for 40 minutes each time and storytelling once a week for the corresponding same amount of time. Are they performing unstructured exercises or other structured exercises? Or are they not practicing?

(3) In 2.9.3, it is recommended that the accelerometer cut points be supplemented. Different cut points affect the proportion of time spent in different activity intensity categories.

(4) 2.11 and 2.12 are recommended to be combined.

Thirdly, the limitation of the study is that it is a prototype study, a pre-determined approach to the design and implementation of the trial, and does not elaborate on the results of the randomized controlled trial. I found out that this study was approved by the ethical review board in 2021, which is now 2 years ago, and if your team has conducted a randomized controlled trial, I would strongly prefer that the article add the results of the trial and a discussion of the results, which I feel would make the article more meaningful.

Fourth, in the discussion section, it is suggested to add a discussion of moderating variables, such as effective time spent with parents, time spent exercising with parents, and whether or not they have participated in out-of-school childcare facilities. Do changes in these variables affect the results of the experiment? Is it necessary to include covariates in tests of intervention effects?

Reviewer 2 Report

The presented study (Protocol for a Randomized Controlled Trialis) focused on the analysis and comparison of the effects of combined movement and storytelling intervention on fundamental motor skills, language development, and physical activity levels. To this end, an experiment will be carried out using a randomized controlled trial, a double-blind study, repeated measurements and parallel groups and a quantitative approach. The study put forward a preliminary hypothesis that the set of special activities proposed by the authors with children has the potential to significantly improve their motor skills, object control, language understanding, language expression, vocabulary, language description and level of physical activity from moderate to high intensity. All these data should grow in the experimental sample, compared with the data of the control group. In the conclusions, the authors point out that, if the experimental intervention is proven to be effective, it could lead to a holistic improvement in the children's curriculum and meet the indicative performance standards associated with preschool organizations for the education of children aged 3 to 6 years.

Despite the fact that the Study Protocol for a Randomized Controlled Trial is structured in detail and shows how the experiment will be carried out in the future, the issue of achieving a positive result is not sufficiently worked out in the review of the study and discussions. The authors also report that there are no known published studies with Hispanic preschoolers that use the same techniques. This, of course, is a great contribution to scientific knowledge in this area of research, but the authors need to refer to the experience of other continents. A review of European and North American studies on such an experimental intervention with children 3 to 6 years of age should be reviewed. Also, the study would have received a higher rating if preliminary results or their comparative detailed analogues for each of the diagnostic areas were shown.

Reviewer 3 Report

I would like to begin by commending the authors for presenting an interesting Study Protocol. This research plan has potential but would benefit from further clarification and elaboration in several areas. I look forward to seeing a revised version of the plan that addresses these comments and suggestions.

1. The abstract could benefit from further clarification. Specifically, it would be helpful to elucidate the relationships among the upper middle level classes, transition level 1 classes, and transition level 2 classes.

2. In the introduction, the variable targeted by reference 4 in terms of the practice of physical activity is not explicitly stated. Is it LPA, MPA, VPA,or  MVPA? Additionally, the correlation coefficient r=0.25 is mentioned, but it is unclear which two variables this refers to. Could the authors please provide more precise explanations? Similarly, the OR in reference 9 is not clearly defined. I kindly request that the authors check the original reference and provide further clarification.

3. In the Sample size calculation section, it is not immediately apparent why only locomotor skills are used for this calculation. Could the authors provide some rationale for this choice?

4. I noticed that the TGMD-2 is used to test FMS instead of the TGMD-3. While the use of various tools is acceptable, it would be helpful to understand the reasoning behind this particular choice.

5. The activities of the control group seem to be a critical factor in this study. Merely being asked to maintain their regular session of their educational establishments may not be sufficient. If the control group participates in Art and craft activities or Outdoor activities and games while the intervention group undergoes CMSI, the effects on the control group's physical and mental development may differ. I kindly suggest that the authors consider setting up different experimental and control groups to provide more targeted improvement suggestions for kindergarten activity arrangements.

6. I noticed that the format of references 4 and 12 appears to be incomplete, as the journal names are missing. I kindly request that the authors check the format of other references according to the requirements of this journal.

     I hope these comments and suggestions will be helpful in improving the research plan.

Reviewer 4 Report

Did you register the clinical trial registry? 

The protocol can be accepted with minor revision. 

Minor editing. 

Reviewer 5 Report

The introduction and discussion paragraphs are too long. Because they involve different messages, they make it difficult to read the information that the authors intend to share. I recommend reviewing and reorganizing the ideas, in shorter paragraphs, strengthening the more objective perspective of the report.

About the selection process, I indicate some points that I consider important. (1) How many education establishments are there in the region of Maule? (2) What is the estimate of children between 3 and 6 years old in the region (e.g., to give the reader a better perspective)? (3) How and for what reasons were the educational establishments chosen (was the choice made for convenience, or was there a selection?).

Reviewer 6 Report

Dear Editor, thank you for the opportunity to review this study. 

It is a very interesting study, however it  lacks results and data analysis. Also the introduction is poorly justified as it provides data, when it should contextualise. 

English should be improved

Round 2

Reviewer 1 Report

The revised version is much better, and I hope your team is able to carry out the experiment successfully.

Reviewer 2 Report

The presented study focused on the analysis and comparison of the effects of combined movement and storytelling intervention on fundamental motor skills, language development, and physical activity levels. To this end, an experiment will be carried out using a randomized controlled trial, a double-blind study, repeated measurements and parallel groups and a quantitative approach. The study put forward a preliminary hypothesis that the set of special activities proposed by the authors with children has the potential to significantly improve their motor skills, object control, language understanding, language expression, vocabulary, language description and level of physical activity from moderate to high intensity. All these data should grow in the experimental sample, compared with the data of the control group. In the conclusions, the authors point out that, if the experimental intervention is proven to be effective, it could lead to a holistic improvement in the children's curriculum and meet the indicative performance standards associated with preschool organizations for the education of children aged 3 to 6 years.

The study states that there are no known published studies involving Latin American preschoolers that have used the same methods. This, of course, is a great contribution to scientific knowledge in this area of research, but as a recommendation, the authors should refer to the experience of other continents. Also, the study would have received a higher practical significance if preliminary results or their comparative detailed analogues for each of the diagnostic areas were shown.

Reviewer 3 Report

The authors have made revisions as requested by the reviewers or provided explanations, and I agree with the publication of this manuscript.

Reviewer 6 Report

The article has been improved. It can be published
